# Relationship between Circulating Serpina3g, Matrix Metalloproteinase-9, and Tissue Inhibitor of Metalloproteinase-1 and -2 with Chronic Obstructive Pulmonary Disease Severity

**DOI:** 10.3390/biom9020062

**Published:** 2019-02-13

**Authors:** Pelin Uysal, Hafize Uzun

**Affiliations:** 1Department of Chest Diseases, School of Medicine, Acibadem Mehmet Ali Aydinlar University, Atakent Hospital, 34303 Istanbul, Turkey; 2Department of Biochemistry, Cerrahpasa Faculty of Medicine, Istanbul University-Cerrahpasa, 34098 Istanbul, Turkey; huzun59@hotmail.com

**Keywords:** COPD, serpina3g, matrix metalloproteinase-9, tissue inhibitor of metalloproteinase-1, tissue inhibitor of metalloproteinase-2

## Abstract

Chronic obstructive pulmonary disease (COPD) is influenced by genetic and environmental factors. A protease-antiprotease imbalance has been suggested as a possible pathogenic mechanism for COPD. Here, we examined the relationship between circulating serpina3g, matrix metalloproteinase-9 (MMP-9), and tissue inhibitor of metalloproteinase-1 and -2 (TIMP-1 and -2, respectively) and severity of COPD. We included 150 stable COPD patients and 35 control subjects in the study. The COPD patients were classified into four groups (I, II, III, and IV), according to the Global Initiative for Chronic Obstructive Lung Disease (GOLD) guidelines based on the severity of symptoms and the exacerbation risk. Plasma serpina3g, MMP-9, and TIMP-1 and -2 concentrations were significantly higher in the all patients than in control subjects. Plasma serpina3g, MMP-9, and TIMP-1 and -2 concentrations were significantly higher in groups III and IV than in groups I and II. A negative correlation between serpina3g, MMP-9, and TIMP-1 and -2 levels and the forced expiratory volume in 1 s (FEV1) was observed. MMP-9 concentration and the MMP-9/TIMP-1 ratio were higher in patients with emphysema than in other phenotypes (both with *p* < 0.01). The findings of this study suggest that circulating serpina3g, MMP-9, and TIMP-1 and -2 levels may play an important role in airway remodeling in COPD pathogenesis. Disrupted protease-antiprotease imbalance in patients with COPD is related to the presence of airway injury. MMP-9 concentration and the MMP-9/TIMP-1 ratio are the best predictors of emphysema in COPD patients.

## 1. Introduction

Chronic obstructive pulmonary disease (COPD) is one of today’s most important global health problems. The increasing prevalence, morbidity, and mortality of the disease are both economic and social burdens [1]. COPD is influenced by genetic and environmental factors. Unexplained genetic factors, namely epigenetic interactions, may lead to the pathogenesis of COPD [2]. In order to prevent the parenchymal inflammation of the lungs secondary to smoking, antiproteases prevent proteolytic activity. As a result of some genetic polymorphisms along with environmental factors, COPD has been suggested to occur as a result of antiprotease ineffectiveness [3].

Alpha-1-antichymotrypsin (AACT), also called serine protease inhibitor (*serpin*) A3 gene (*serpina3g*), is an alpha globulin glycoprotein that is a member of the serpin (also referred to as antiprotease) family of acute phase proteins, which includes 36 human proteins. According to the terminology for serpins, the human genome encodes 16 guides from serpin A to P of the sprinkles, and the corresponding genes include 29 inhibitory and 7 non-inhibitory genes [3,4]. Several studies have reported that serpina1, serpina2, and serpina3 are associated with COPD [5,6], Serpina3-encoded AACT is a plasma protease inhibitor that targets neutrophil cathepsin G and elastase [7,8]. Although mutations in serpina3g have been shown to be related to COPD in some studies, a similar but less obvious relationship was found between the common variants of COPD and serpina3g [9,10].

In recent years, the deterioration of the protease-antiprotease balance has been suggested to play a role in the pathogenesis of COPD. Matrix metalloproteinases (MMPs), which affect this equilibrium, are indispensable elements of tissue reconstruction [11,12]. MMPs released from macrophages lead to the destruction of elastin and cause emphysema [13]. These enzymes participate in normal body functions but they behave differently in COPD [14,15]. MMPs synthesized by inflammatory and structural cells, mainly by macrophages, break the peptide bonds of protein chains [16,17]. Fibroblasts and macrophages also produce endogenous tissue metalloproteinase inhibitors (TIMPs) [15] that selectively inhibit the proteolytic effects of endopeptidases. Different TIMPs have been defined, the most important of which is TIMP-1, which inhibits all active MMPs by creating reversible noncovalent bonds. TIMP-1 was found to effectively inhibit both the active and inactive first form of MMP-9 [18,19]. The genetic polymorphisms of TIMP-1 and TIPM-2 are the tissue inhibitors of MMPs that have been found to be associated with COPD [13].

COPD pathogenesis may be caused by a dysregulation of protease activity when an imbalance between proteases and antiproteases develops because of either the higher activity of proteases or dysfunction of protease inhibitors [20,21,22,23].

It has been hypothesized that protease-antiprotease imbalance occurs in patients with COPD. For this reason, in this study, we evaluated the relationship between the levels of circulating serpina 3g, MMP-9, TIMP-1, -2, MMP-9/TIMP-1, and MMP-9/TIMP-2 ratio with clinical features of COPD.

## 2. Materials and Methods

The present study was approved by the ethics committee of Cerrahpasa Medical Faculty, Istanbul University and was conducted in accordance with the Declaration of Helsinki. Written informed consent was provided by all the participating subjects (ethics committee approval ID: 2016/6781). This prospective study was performed at Acibadem Mehmet Ali Aydinlar University, School of Medicine, Department of Chest Disease. We enrolled 150 stable COPD patients and 35 controls in this study. The control subjects were selected from a group of nonsmoker healthy persons who presented at the Department of Chest Diseases for regular health control or check-ups. The spirometry test was performed on the control subjects.

A detailed history was taken from all participants and physical examinations were performed. Patients with respiratory disorders (pulmonary embolism, left ventricular systolic or diastolic dysfunction) other than COPD, comorbidities such as diabetes, chronic renal insufficiency, dysthyroidism, hepatic dysfunction, lower respiratory tract infection, COPD attack in the last six weeks, and presence of metabolic syndrome were excluded from the study. COPD and its comorbidities were followed for all patients, and a heart echocardiography was performed as well.

Diagnosis of COPD was established according to spirometry results, i.e., expiratory volume in one second (FEV1)/forced vital capacity (FVC) ratio < 70%, according to the 2017 Global Initiative for Chronic Obstructive Lung Disease (GOLD) guidelines [24]. Exacerbations of COPD were graded as levels I–IV according to the European Respiratory Society (ERS)/American Thoracic Society (ATS) consensus criteria [25]. All patients were managed in accordance with ERS/ATS guidelines, including bronchodilators and inhaler corticosteroids.

Spirometry tests were performed in accordance with the criteria recommended by the European Respiratory Society using computer-assisted spirometry (Vmax22D, Sensor Medics, Yorba Linda, CA, USA). Pulmonary function parameters, such as FEV1, FVC, and FEV1/FVC ratio, were measured and the absolute values and the percentage of expected values of these parameters were analyzed. These patients had a stable airflow limitation with FEV1 < 80% of the predicted value in combination with a FEV1/FVC ratio < 70% predicted, with a reversibility of < 12% predicted postbronchodilator. The COPD groups were classified as mild, moderate, severe, or very severe according to GOLD: mild COPD (FEV1 ≥ 80% predicted) with FEV1/FVC < 0.70; moderate COPD (50% ≤ FEV1 < 80%) with FEV1/FVC < 0.70; severe COPD (30% ≤ FEV1 < 50%) with FEV1/FVC < 0.70; and very severe COPD (FEV1 < 30% predicted) with FEV1/FVC < 0.70.

Regarding the distribution by phenotype, there were 28 patients (20%) with asthma-COPD overlap (ACO), 63 (45%) with chronic bronchitis (CB), and 49 (35%) with emphysema. Patients who had COPD, asthma, or ACO, according to the ACO 2016 guideline, and had a moderate obstruction (50% < FEV1 < 80%) in pulmonary function tests (PFTs) were included in the study [26]. Chronic bronchitis is clinically defined by the presence of cough and sputum production for at least three months in each of two consecutive years.

### 2.1. Measurement of Emphysema by High-Resolution Computed Tomography Scan

All patients underwent lung high-resolution computed tomography (HRCT) performed at full inspiration and full expiration. A subjective, semiquantitative measurement of emphysema was performed by visual assessment of HRCT images using Syngo CT SOMATOM Definition Flash Pulmo 3D image processing software (Siemens AG, Forchheim, Germany) with a low attenuation area (LAA%) defined as less than −960 HU.

### 2.2. Laboratory Analysis

#### 2.2.1. Sample Collection and Preparation

Venous blood samples were collected in the morning in tubes containing ethylenediaminetetraacetic acid (EDTA) and anticoagulant-free tubes after 12 h of fasting and before drug use. Plasma and serum samples were separated by centrifugation at 3000 × *g* for 10 min at 4°C and then separated in microcentrifuge tubes and frozen immediately at −80 °C until analysis.

Routine biochemical parameters were measured by an autoanalyzer (Hitachi Modular System, Roche Diagnostic, Indianapolis, USA). Serum C-reactive protein (CRP) levels were measured by a nephelometric method (Immage 800 Beckman Coulter, Fullerton, CA, USA). Complete blood count parameters were obtained with an automatic hematology analyzer (Siemens-Sysmex, Eschborn, Germany). Erythrocyte sedimentation rate (ESR) was measured according to the Westergren method with an established normal range (0–20 mm/h).

#### 2.2.2. Measurement of Plasma Serpina3g Concentration

Plasma serpina3g concentration was measured using a solid-phase enzyme-linked immunosorbent assay (ELISA) kit based on the sandwich principle (Human Serine Protease Inhibitor A 3G ELISA Kit, Cat. No. E1874Hu, Bioassay Technology Laboratory, Shanghai, China). Calculated intra- and inter-assay coefficients of variation (CVs) were 7.8% (*n* = 15) and 8.9% (*n* = 15) for this kit, respectively.

#### 2.2.3. Measurement of MMP-9 Concentration

Plasma MMP-9 concentration was measured by a solid-phase ELISA kit based on the sandwich principle (Human Matrix Metalloproteinase-9 ELISA Kit, Cat. No. E0936Hu, Bioassay Technology Laboratory, Shanghai, China). Calculated intra- and inter-assay CVs were 7.9% (*n* = 15) and 8.9% (*n* = 15) for this kit, respectively.

#### 2.2.4. Measurement of TIMP-1 Concentration

Plasma TIMP-1 concentration was measured by a solid-phase ELISA kit based on the sandwich principle (Human Tissue Inhibitors of Metalloproteinase-1 ELISA Kit, Cat. No. E1236Hu, Bioassay Technology Laboratory, Shanghai, China). Calculated intra- and inter-assay CVs were 8.1% (*n* = 15) and 9.1% (*n* = 15) for this kit, respectively.

#### 2.2.5. Measurement of TIMP-2 Concentration

Plasma TIMP-2 concentration was measured by a solid-phase ELISA kit based on the sandwich principle (Human Tissue Inhibitors of Metalloproteinase-2 ELISA Kit, Cat. No. E1218Hu, Bioassay Technology Laboratory, Shanghai, China). Calculated intra- and inter-assay CVs were 7.9% (*n* = 15) and 8.9% (*n* = 15) for this kit, respectively.

### 2.3. Statistical Analysis

Statistical analyses were performed using SPSS 20.0 (SPSS Inc., Chicago, IL, USA). All data were first checked for normality. Normally distributed continuous variables are presented as mean ± standard deviation (SD) and were analyzed by one-way analysis of variance (ANOVA) followed by Tukey’s multiple comparison tests. Pearson and Spearman’s correlations were used for numerical and nominal data, respectively.

## 3. Results

The overall characteristics of the groups are summarized in Table 1. As expected, groups I–IV had significantly lower FEV1 and FEVC than controls (*p* < 0.001 for all subgroups). The lowest FEV1 and FEV1/FVC levels were obtained from group IV.

The protease-antiprotease levels of the groups are presented in Figure 1. Plasma serpina3g concentration was found to be significantly increased in patients with COPD, especially in group IV, when compared with the other groups (*p* < 0.001). Our results indicated that MMP-9 concentration increased as follows: control < group I < group II < group III < group IV. Their levels of significance compared to the control group, respectively, were *p* < 0.01, *p* < 0.01, *p* < 0.001, and *p* < 0.001. Also, when we compared COPD groups among themselves, higher TIMP-1 and -2 concentrations were found in groups III and IV than in groups I and II (each with *p* < 0.001). No statistically significant difference was found between plasma serpina3g, MMP-9, and TIMP-1 and -2 concentrations in groups I and II. Also, no statistically significant difference was found for MMP-9/TIMP-1 in all groups. A higher MMP-9/TIMP-2 ratio was found in group III compared with the control (*p* < 0.01) and group I (*p* < 0.05). The MMP-9/TIMP-2 ratio was higher in group IV than in the control (*p* < 0.001) and group I (*p* < 0.01). No statistically significant difference was found for the MMP-9/TIMP-2 ratios in groups II–IV.

MMP-9 concentrations and the MMP-9/TIMP-1 ratio were higher in patients with emphysema than in other phenotypes (both *p* < 0.01). Also, there was no difference in the other parameters between phenotypes. FEV1 was correlated with MMP-9/TIMP-1 imbalance in patients with emphysema (*r* = 0.664, *p* < 0.001).

The correlation analyses of group IV and all subjects are presented in Table 2 and Table 3, respectively. 

## 4. Discussion

The main finding of this study was that the disrupted protease-imbalance tends to be elevated during the disease, with decreased PFTs in COPD patients. An inverse correlation was observed between TIMP-1 and -2, serpina3g, and MMP-9 concentration with changes in FEV1 and FEV1/FVC in group IV and all COPD patients, which may be useful for predicting COPD activity and patient prognosis. In our study, MMP-9 concentration and the MMP-9/TIMP-1 ratio were the best predictors of emphysema in COPD patients.

COPD and its attendant complications pose a significant public health problem. This disease is a chronic process that includes inflammation and remodeling. Although many studies have been conducted on genetic factors, the most well-known genetic risk factor for COPD formation is AAT deficiency. However, severe AAT deficiency plays a role as a risk factor in only 1% of patients with COPD [27]. Therefore, the role of genetic factors in the occurrence of COPD is one of the most popular research areas. Genetic risk factors that are responsible for the occurrence of COPD are gene disorders that regulate the protease and antiprotease enzymes. The most important gene disorders that regulate protease and antiprotease enzymes are AAT deficiency, serpins, α2-macroglobulin, and AACT polymorphisms [2]. AAT is an antiprotease that inhibits the destructive effect of neutrophil elastase on lung tissue by irreversibly inhibiting serine proteases (neutrophil elastase, cathepsin G, and proteinase) [28]. MMPs released from macrophages lead to the destruction of elastin and emphysema [29].

Extracellular matrix proteins promote proliferation, migration, and adhesion of airway smooth muscle cells in COPD [30]. Proteolytic biomarkers were found to be related to the prognosis of COPD in a population-based study [31]. MMP-9 in exhaled breath condensates in patients with stable COPD [32]. MMP-9 has a significant positive correlation with the degree of inflammatory metalloproteinase cell infiltration in COPD patients [33]. Genetic polymorphism in MMP-9 and TGF-β1 have been found to be associated with pulmonary fibrosis and emphysema in a Chinese population [34]. Also, MMP-8, MMP-9, and neutrophil elastase in peripheral blood were found to be higher in COPD patients [35]. In our study, plasma MMP-9 concentration was found to be lower in controls and increased as the disease progressed. MMP-9 concentration and the MMP-9/TIMP-1 ratio were higher in patients with emphysema than in other phenotypes, and FEV1 was correlated with MMP-9/TIMP-1 imbalance. These results show one potential mechanism for the formation of emphysema with COPD. Our findings are consistent with the literature [36].

In humans, several conformational diseases, or serpinopathies linked to serpin polymerization, have been identified, including emphysema serpin A1 (AAT deficiency), thrombosis serpin C1 (antithrombin) deficiency, and angioedema serpin G1 (C1 esterase inhibitor) deficiency [6]. The most widely recognized candidate gene in COPD is *serpina1*, although it has been suggested that *serpina3g* may also play a role [37]. Ishii et al. [8] suggested that genetic polymorphism in the signal peptide of AACT may be associated with individual susceptibility to the development of COPD because the AACT/Ala-15 genotype is predominantly found in patients with COPD. Hollander et al. [38] suggested that increased plasma levels of serine protease inhibitors, such as ACT, might favorably improve the protease-antiprotease balance in COPD patients with severe AAT deficiency. Our aim was to identify serpina3g levels in COPD patients. In our study, serpina3g levels were lower in the control group than in COPD patients. We found that the serpina3g levels increased as the disease progressed and are positively correlated with MMP-9 levels.

TIMP-1 is a natural inhibitor of MMPs and a glycoprotein expressed in multiple tissues in many organisms [39]. In addition to its inhibitory role, TIMP-1 is able to promote cell proliferation and inhibit apoptosis as well as regulate cell growth in a wide range of cell types [40]. Evidence suggests that MMP-9 is inhibited by TIMP-1, and an imbalance in the MMP-9/TIMP-1 ratio could be involved in COPD pathogenesis, although conflicting results have been reported [41,42,43]. Mutations of TIMP-2 downregulate its activity and may increase the activities of matrix metalloproteinases, resulting in the degradation of the lung matrix [44]. TIMP-1 and -2 have been shown to participate in pulmonary diseases characterized by alterations of the alveolar structure or abnormal remodeling responses such as emphysema, interstitial fibrosis, acute respiratory distress syndrome, and lung cancer [44]. Fujita et al. [45] found that the activities of both MMP-9 and TIMP-1 following lung injury in COPD led to either recovery or degradation of the extracellular matrix (ECM). Among smokers without COPD, the activities of MMP-9 and TIMP-1 result in recovery and resolution, whereas for smokers with COPD, this results in ECM destruction and emphysema. However, the preliminary findings of Mulyadi et al. [46] suggested that salivary TIMP-1 is not a suitable biomarker in Indonesian subjects with COPD. In our study, TIMP-1 and -2 levels were lower in the control group but were higher in patients with severe COPD. These increases in TIMP-1 and -2 may be compensatory increases. TIMP-1 has been found to inhibit MMP-9 by binding to its precursors and active forms. Alveolar macrophages are a significant source of MMP-9 that triggers the formation of larger amounts of MMP-9 in COPD patients. The increased MMP-9 might be due to the relative decrease in the release of the specific endogenous inhibitor of MMP-9, TIMP-1. The imbalance between the levels of MMP-9 and TIMP-1 might result in aberrant ECM degradation or the accumulation of ECM proteins in pulmonary alveoli and small airway walls, which could lead to COPD [36]. Further studies with a more heterogeneous population are required to clarify the precise role of TIMPs as a possible biomarker of antiprotease activity in the lung [46]. 

However, this study has some limitations. First, our sample size was relatively small. Second, the physical activity and exercise levels of the subjects were not documented. The levels of proteases and antiproteases were analyzed, not activities. COPD has been shown to be associated with genetic factors. However, a genetic analysis was not completed in these COPD patients. We could not measure these parameters in sputum and bronchoalveolar lavage (BAL) specimens.

In severe groups of patients (groups III and IV), plasma serpina3g and MMP-9 concentration had the highest values, and TIMP-1 and -2 were higher in groups III and IV. These increases in TIMP-1 and -2 may be compensatory increases. We think that TIMP-1 and -2 levels increased as a compensatory mechanism to suppress the MMP-9 increase. Since measurements were recorded by sampling blood at a single time, TIMP-1 and -2 levels may be measured at the time when they increase as the compensatory mechanism. If measurements were recorded at different time intervals, we could show a decrease in TIMP-1 and -2 levels. When we examined the relationship of disease characteristics and plasma serpina3g, MMP-9, and TIMP levels, we found a negative correlation with FEV1 and serpina3g and MMP-9. Thus, increased circulating MMP-9 and serpina3g in patients with COPD was related to poor lung function and the presence of airway injury. MMP-9 concentration and the MMP-9/TIMP-1 ratio were the best predictors of emphysema in COPD patients. Circulating serpina3g, MMP-9, and TIMP-1 and -2 may also act in COPD pathogenesis. The role of protease-antiprotease imbalance should be further studied to improve our understanding of COPD pathogenesis and progression.

## Figures and Tables

**Figure 1 biomolecules-09-00062-f001:**
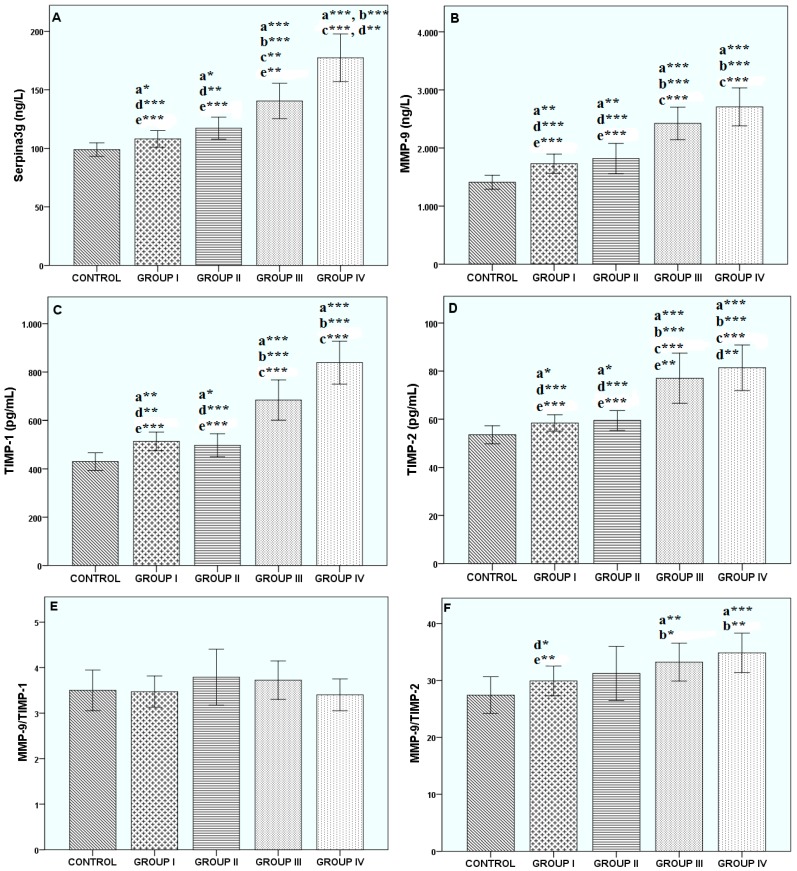
The changes in plasma (**A**) serpina3g, (**B**) matrix metalloproteinase-9 (MMP-9), (**C**) tissue inhibitor of metalloproteinase-1 (TIMP-1), (**D**) tissue inhibitor of metalloproteinase-2 (TIMP-2), (**E**) MMP-9/TIMP-1, and (**F**) MMP-9/TIMP-2 of all subjects. ^*^*p* < 0.05, ^**^*p* < 0.01, ^***^*p* < 0.001. ^a^ vs. Control, ^b^ vs. group I, ^c^ vs. group II, ^d^ vs. group III, ^e^ vs. group IV.

**Table 1 biomolecules-09-00062-t001:** Demographic, clinical, and laboratory findings of all groups.

	Control (*n* = 35)	Group I (Mild) (*n* = 45)	Group II (Moderate) (*n* = 35)	Group III (Severe) (*n* = 35)	Group IV (Very Severe) (*n* = 35)
**Age (years)**	46.91 ± 7.96	57.78 ± 11.88 ^a***,c*^	63.31 ±11.98 ^a***, b*^	67.03 ± 11.13 ^a***, b***^	65.44 ± 10.45 ^a***, b**^
**Female/Male**	15/20	16/34	13/19	10/20	11/27
**FEV1** **(% predicted)**	102.06 ± 8.40	78.84 ± 15.25 ^a***,d***,e***^	75.50 ± 10.50 ^a***,d***,e***^	40.33 ± 9.36 ^a***,b***,c***,e*^	35.00 ± 10.17 ^a***,b***,c***,d*^
**FEV1/FVC**	83.91 ± 4.34	65.31 ± 7.16 ^a***,d***,e***^	67.41 ± 3.77 ^a***,d***,e***^	55.20 ± 10.04 ^a***,b***,c***,e*^	49.54 ± 9.90 ^a***,b***,c***,d*^
**WBC (×10^3^/μL)**	7.43 ± 1.19	8.30 ± 3.03	7,59 ± 2,07 ^e*^	8,97 ± 3.51	9.14 ± 2.99 ^a***,c*^
**Total Protein (g/dL)**	8.06 ± 0.36	7.38 ± 0.48 ^a***,e***^	7,31 ± 0,60 ^a***,e***^	7,28 ± 0,45 ^a***,e**^	6.91 ± 0,64 ^a***,b***,c***,d**^
**Albumin (g/dL)**	4.06 ± 0.28	3.62 ± 0.38 ^a***,e**^	3,66 ± 0,43 ^a***,e***^	3,47 ± 0,44 ^a***^	3.39 ± 0.50 ^a***,b**,c***^
**ESR (mm/h)**	11.47 ± 5.90	18.18 ± 11.38 ^a***^	18.73 ± 11.13 ^a***^	22.30 ± 12.59 ^a***^	21.95 ± 15.83 ^a***^
**CRP (mg/L)**	0.33 ± 0.19	0.65 ± 0.55 ^a***,d*^	0.52 ± 0.50	0.74 ± 0.61 ^a***^	0.58 ± 0.50 ^a**^

Note: FEV1: forced expiratory volume in the first second, FVC: forced vital capacity, WBC: white blood cell, ESR: erythrocyte sedimentation rate, CRP: C-reactive protein. ^*^*p* < 0.05, ^**^*p* < 0.01, ^***^*p* < 0.001. ^a^ vs. Control, ^b^ vs. Group A, ^c^ vs. Group B, ^d^ vs. Group C, ^e^ vs. Group D.

**Table 2 biomolecules-09-00062-t002:** Correlation analysis between protease–antiprotease levels and FEV1 and FEV1/FVC in group IV.

		Serpina3g (ng/mL)	MMP-9 (ng/mL)	TIMP-1 (pg/mL)	TIMP-2 (pg/mL)	MMP-9/TIMP-1	MMP-9/TIMP-2	FEV1 (% Predicted)	FEV1/FVC
**Serpina3g** **(ng/mL)**	**r**	1	0.576 ^**^	0.734 ^**^	0.839 ^**^	−0.197	−0.259	−0.666 ^**^	−0.431 ^**^
	**p**		0.000	0.000	0.000	0.230	0.111	0.000	0.006
**MMP-9** **(ng/mL)**	**r**	0.576 ^**^	1	0.578 ^**^	0.627 ^**^	0.384 ^*^	0.426 ^**^	−0.477 ^**^	−0.305
	**p**	0.000		0.000	0.000	0.016	0.007	0.002	0.059
**TIMP-1** **(ng/mL)**	**r**	0.734 ^**^	0.578 ^**^	1	0.773 ^**^	−0.508 ^**^	−0.212	−0.934 ^**^	−0.699 ^**^
	**p**	0.000	0.000		0.000	0.001	0.195	0.000	0.000
**TIMP-2**	**r**	0.839 ^**^	0.627 ^**^	0.773 ^**^	1	−0.202	−0.416 ^**^	−0.656 ^**^	−0.423 ^**^
	**p**	0.000	0.000	0.000		0.218	0.008	0.000	0.007
**MMP-9/TIMP-1**	**r**	−0.197	0.384 ^*^	−0.508 ^**^	−0.202	1	0.670 ^**^	0.553 ^**^	0.507 ^**^
	**p**	0.230	0.016	0.001	0.218		0.000	0.000	0.001
**MMP-9/TIMP-2**	**r**	−0.259	0.426 ^**^	−0.212	−0.416 ^**^	0.670 ^**^	1	0.192	0.117
	**p**	0.111	0.007	0.195	0.008	0.000		0.241	0.479
**FEV1** **(% predicted)**	**r**	−0.666 ^**^	−0.477 ^**^	−0.934 ^**^	−0.656 ^**^	0.553 ^**^	0.192	1	0.741 ^**^
	**p**	0.000	0.002	0.000	0.000	0.000	0.241		0.000
**FEV1/FVC**	**r**	−0.431 ^**^	−0.305	−0.699 ^**^	−0.423 ^**^	0.507 ^**^	0.117	0.741 ^**^	1
	**p**	0.006	0.059	0.000	0.007	0.001	0.479	0.000	

Note: MMP-9: matrix metalloproteinase-9, TIMP-1: tissue inhibitor of metalloproteinase-1, FEV1: forced expiratory volume in the first second, FVC: forced vital capacity. ** Correlation is significant at the 0.01 level (two-tailed); * Correlation is significant at the 0.05 level (two-tailed).

**Table 3 biomolecules-09-00062-t003:** Correlation analysis between protease–antiprotease levels and FEV1 and FEV1/FVC in all patients.

		Serpina3g (ng/mL)	MMP-9 (ng/mL)	TIMP-1 (pg/mL)	TIMP-2 (pg/mL)	MMP-9/TIMP-1	MMP-9/TIMP-2	FEV1 (% Predicted)	FEV1/FVC
**Serpina3g** **(ng/mL)**	**r**	1	0.597 ^**^	0.719 ^**^	0.674 ^**^	−0.062	0.047	−0.625 ^**^	−0.567 ^**^
	**p**		0.000	0.000	0.000	0.445	0.566	0.000	0.000
**MMP-9** **(ng/mL)**	**r**	0.597 ^**^	1	0.628 ^**^	0.629 ^**^	0.456 ^**^	0.587 ^**^	−0.583 ^**^	−0.467 ^**^
	**p**	0.000		0.000	0.000	0.000	0.000	0.000	0.000
**TIMP-1** **(ng/mL)**	**r**	0.719 ^**^	0.628 ^**^	1	0.755 ^**^	−0.341 ^**^	0.021	−0.691 ^**^	−0.684 ^**^
	**p**	0.000	0.000		0.000	0.000	0.801	0.000	0.000
**TIMP-2**	**r**	0.674 ^**^	0.629 ^**^	0.755 ^**^	1	−0.103	−0.217 ^**^	−0.539 ^**^	−0.526 ^**^
	**p**	0.000	0.000	0.000		0.208	0.007	0.000	0.000
**MMP-9/TIMP-1**	**r**	−0.062	0.456 ^**^	−0.341 ^**^	−0.103	1	0.695 ^**^	0.000	0.163 ^*^
	**p**	0.445	0.000	0.000	0.208		0.000	0.996	0.045
**MMP-9/TIMP-2**	**r**	0.047	0.587 ^**^	0.021	−0.217 ^**^	0.695 ^**^	1	−0.221 ^**^	−0.055
	**p**	0.566	0.000	0.801	0.007	0.000		0.006	0.504
**FEV1** **(% predicted)**	**r**	−0.625 ^**^	−0.583 ^**^	−0.691 ^**^	−0.539 ^**^	0.000	−0.221 ^**^	1	0.731 ^**^
	**p**	0.000	0.000	0.000	0.000	0.996	0.006		0.000
**FEV1/FVC**	**r**	−0.567 ^**^	−0.467 ^**^	−0.684 ^**^	−0.526 ^**^	0.163 ^*^	−0.055	0.731 ^**^	1
	**p**	0.000	0.000	0.000	0.000	0.045	0.504	0.000	

Note: MMP-9: matrix metalloproteinase-9, TIMP-1: tissue inhibitor of metalloproteinase-1, FEV1: forced expiratory volume in the first second, FVC: forced vital capacity. ** Correlation is significant at the 0.01 level (two-tailed); * Correlation is significant at the 0.05 level (two-tailed).

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
