# Peer review of "Relationship between Circulating Serpina3g, Matrix Metalloproteinase-9, and Tissue Inhibitor of Metalloproteinase-1 and -2 with Chronic Obstructive Pulmonary Disease Severity"

_biomolecules, 2019, doi:10.3390/biom9020062_

Round 1
Reviewer 1 Report
The authors have adequately addressed the issues in their revision. I just have a couple of minor comments:
1) Page 2 line 64 - Replace 'COPD pathogenesis is dysregulation of proteases activity, especially when imbalance...' with 'COPD pathogenesis may be caused by a dysregulation of protease activity, when an imbalance...'
2) Figure 1 - the use of letters and numbers to indicate significance is between groups is difficult to read without zooming in on the text. The results would be much clearer if lines and the standard symbols (*, **, ***) to indicate significant differences between groups were employed.
3) Page 9 lines 254-255 - The meaning behind the sentence 'Increased expression of MMP-9 and TIMP-1 and alveolar macrophages trigger larger amounts of MMP-9 with greater enzymatic activity in COPD patients.' is unclear and requires further clarification.
4) Page 9 lines 261-262 - Revise the sentence 'The levels of these proteins were analyzed, not activities.' to read 'The levels of proteases and antiproteases were analyzed, not activities.'
Author Response
Dear Editor,
First, we would like to acknowledge the thoughtful comments of the reviewers, which led us to conduct appropriate experiments. The manuscript has subsequently been rewritten to address the concerns and comments of the reviewers.
We are grateful for your understanding and cooperation in this matter.
Response to Reviewers:
Thank you for valuable comments.
Reviewer 1
Open Review
(x) I would not like to sign my review report
( ) I would like to sign my review report
English language and style
( ) Extensive editing of English language and style required
( ) Moderate English changes required
(x) English language and style are fine/minor spell check required
( ) I don't feel qualified to judge about the English language and style
Yes | Can be improved | Must be improved | Not applicable | |
Does the introduction provide sufficient background and include all relevant references? | (x) | ( ) | ( ) | ( ) |
Is the research design appropriate? | (x) | ( ) | ( ) | ( ) |
Are the methods adequately described? | (x) | ( ) | ( ) | ( ) |
Are the results clearly presented? | ( ) | (x) | ( ) | ( ) |
Are the conclusions supported by the results? | (x) | ( ) | ( ) | ( ) |
Comments and Suggestions for Authors
The authors have adequately addressed the issues in their revision. I just have a couple of minor comments:
1) Page 2 line 64 - Replace 'COPD pathogenesis is dysregulation of proteases activity, especially when imbalance...' with 'COPD pathogenesis may be caused by a dysregulation of protease activity, when an imbalance...'
Sentences were corrected.
2) Figure 1 - the use of letters and numbers to indicate significance is between groups is difficult to read without zooming in on the text. The results would be much clearer if lines and the standard symbols (*, **, ***) to indicate significant differences between groups were employed.
We made the necessary changes.
3) Page 9 lines 254-255 - The meaning behind the sentence 'Increased expression of MMP-9 and TIMP-1 and alveolar macrophages trigger larger amounts of MMP-9 with greater enzymatic activity in COPD patients.' is unclear and requires further clarification.
We made the necessary changes.
4) Page 9 lines 261-262 - Revise the sentence 'The levels of these proteins were analyzed, not activities.' to read 'The levels of proteases and antiproteases were analyzed, not activities.'
We made the necessary changes.

Reviewer 2 Report
The new version of the manuscript greatly improved the quality of the study. The authors considered the suggested comments and reanalysed data in a different view. The new results confirmed that biomarkers like MMP-9 and TIMP-1, TIPM-2 and serpina3g may be helpful in distinguishing various COPD phenotypes.
Minor remarks:
1. Control group characteristic: Smoking status and spirometry tests should be included into the text.
2. I would suggest to change the aim of the study:
For this reason, in this study, we evaluated the relationship between level of circulating serpina 3g, MMP-9, TIMP-1, -2, MMP-9/TIMP-1, and MMP-9/TIMP-2 ratio with clinical features of COPD .
3. Authors added the detailed diagnosis criteria for CODP, among others management of patients where oral corticosteroids and antibiotics are described at the same tame they wrote that lower respiratory tract infection and COPD attack in the last 6 weeks were excluded from the study. Please specify if patients with this treatment were included into this study.
4. According to GINA 2018 : “in order to avoid the impression that this is a single disease, the term Asthma COPD Overlap Syndrome (ACOS), used in previous versions of this document, is no longer advised.” I suggest to replace ACOS with ACO
5. How were patients with chronic bronchitis characterized?
6. I suggest to replace Table 2 with figure or table for the distribution of MMP9/TIMP1 and MMP9/TIMP2 ratio in ACO, chronic bronchitis and emphysema.
7. The statistical analysis for FEV1 were made for % of predicted or for the liters? Please specify the units for FEV1.
8. Please specify the plasma serpina3g, MMP-9, and TIMP-1 and -2 concentrations units in table 2 (if stays) and table 3.
Author Response
Dear Editor,
First, we would like to acknowledge the thoughtful comments of the reviewers, which led us to conduct appropriate experiments. The manuscript has subsequently been rewritten to address the concerns and comments of the reviewers.
We are grateful for your understanding and cooperation in this matter.
Response to Reviewers:
Thank you for valuable comments.
Reviewer 2
Open Review
(x) I would not like to sign my review report
( ) I would like to sign my review report
English language and style
( ) Extensive editing of English language and style required
( ) Moderate English changes required
(x) English language and style are fine/minor spell check required
( ) I don't feel qualified to judge about the English language and style
Yes | Can be improved | Must be improved | Not applicable | |
Does the introduction provide sufficient background and include all relevant references? | (x) | ( ) | ( ) | ( ) |
Is the research design appropriate? | (x) | ( ) | ( ) | ( ) |
Are the methods adequately described? | ( ) | (x) | ( ) | ( ) |
Are the results clearly presented? | ( ) | (x) | ( ) | ( ) |
Are the conclusions supported by the results? | (x) | ( ) | ( ) | ( ) |
Comments and Suggestions for Authors
The new version of the manuscript greatly improved the quality of the study. The authors considered the suggested comments and reanalysed data in a different view. The new results confirmed that biomarkers like MMP-9 and TIMP-1, TIPM-2 and serpina3g may be helpful in distinguishing various COPD phenotypes.
Minor remarks:
1. Control group characteristic: Smoking status and spirometry tests should be included into the text.
We made the necessary changes.
2. I would suggest changing the aim of the study:
For this reason, in this study, we evaluated the relationship between level of circulating serpina 3g, MMP-9, TIMP-1, -2, MMP-9/TIMP-1, and MMP-9/TIMP-2 ratio with clinical features of COPD.
We made the necessary changes.
3. Authors added the detailed diagnosis criteria for CODP, among others management of patients where oral corticosteroids and antibiotics are described at the same tame they wrote that lower respiratory tract infection and COPD attack in the last 6 weeks were excluded from the study. Please specify if patients with this treatment were included into this study.
We made the necessary changes.
4. According to GINA 2018 : “in order to avoid the impression that this is a single disease, the term Asthma COPD Overlap Syndrome (ACOS), used in previous versions of this document, is no longer advised.” I suggest to replace ACOS with ACO.
We made the necessary changes.
5. How were patients with chronic bronchitis characterized?
CB is defined as the presence of cough and sputum production for at least three months in each of two consecutive years
6. I suggest replacing Table 2 with figure or table for the distribution of MMP9/TIMP1 and MMP9/TIMP2 ratio in ACO, chronic bronchitis and emphysema.
Table 2 was not deleted.
7. The statistical analysis for FEV1 were made for % of predicted or for the liters? Please specify the units for FEV1.
We made the necessary changes.
8. Please specify the plasma serpina3g, MMP-9, and TIMP-1 and -2 concentrations units in table 2 (if stays) and table 3.
We made the necessary change.

This manuscript is a resubmission of an earlier submission. The following is a list of the peer review reports and author responses from that submission.
Round 1
Reviewer 1 Report
The manuscript entitled: Relationship between Circulating Serpina3g, Activity of Matrix Metalloproteinase-9 and Tissue Inhibitor of Metalloproteinase-1, 2 with Chronic Obstructive Pulmonary Disease severity concerns the issue of the role of protease/antiprotease in COPD pathobiology and its correlation with the clinical features of the disease. The study contains several flaws:
Major remarks:
· The study lacks the novelty and the innovation element.
· The methodology described in the manuscript (ELISA kits) assessed the concentration of serpina3g, MMP-9, TIPM-1 and TIMP-2 in serum but not activity. For activity evaluation other assays should be used (for example QuickZyme or innoZyme for MMP9). The term activity should be replaced with level or concentration.
· The authors focused on protease/antiprotease relationship in COPD claimed that AAT deficiency is the well-known genetic factor for this disease but as they say in discussion only in 1% of COPD patients. COPD is a heterogenous disease with multiple described phenotypes. Also describing of novel treatable traits is important path of obstructive lung disease better treatment and more effective clinical control. The known COPD clinical phenotypes are for example emphysema and chronic bronchitis. The detailed characterization of patients with emphysema (using CT) in context of described protease/antiprotease relationship would improve the study.
· The work based on the statistics using singular biomarker level. Did authors tried the analysis with protease/antiprotease concentration ratio, or for example hierarchical clustering with clinical features correlated with emphysema.
· The discussion contains previously described in introduction information, lacks the limitless of the study section and better formed conclusion with the potentially practical application of the presented results.
· The correlations were made for number of exacerbations not the exacerbations in CODP patients. The characteristics of exacerbations for COPD groups its number and period are not shown in the manuscript.
· It is not clear for me on what criteria authors formed the conclusion that “MMP9 is the best predictor of exacerbations in COPD patients”?
Reviewer 2 Report
Thank you for the opportunity to review this manuscript for publication. The authors here identify key proteases and protease inhibitors that may be associated with COPD severity and exacerbation risk. These factors have not previously been investigated with respect to COPD and this study is therefore of interest to COPD researchers and clinicians.
While the study aims and methods are sound, the results could be presented in a more self-explanatory manner. It is difficult to interpret the data in the tables – for example, a graphical format to show error bars and p-values might be more appropriate. It would also greatly improve the manuscript if the correlations were presented in a Figure/Table. There are minor language errors (eg. FEV! Instead of FEV1; “The data of protease and antiprotease presented in Table 2.” – incomplete sentence; “may play an important in role airway remodelling” – should be “role in”). In addition, the manuscript could benefit from an extended discussion – for example, it is difficult to determine the relationship between serpina3g and the MMP and TIMP1/2 and how they collectively contribute to protease/anti-protease imbalance – it could be two separate studies. There also needs to be more discussion regarding the suggestion that the increase in TIMP1/2 is compensatory – if there is an imbalance in MMP and TIMP1/2 is compensating, it is unclear as to why is this enhancing a disease state rather than “balancing” out and returning the body to normal?
Reviewer 3 Report
The authors present analysis of plasma from 150 stable COPD patients and 34 - 35 control subjects. The overall focus of the study was to determine the relationships (as assessed by correlations) between circulating proteases (MMP-9), circulating antiproteases (the serpin AACT, and the MMP inhibitors TIMP-1 and TIMP-2) and severity of disease. The data presented are interesting and they confirm previous research in the field, however I do have a number of concerns.
Major comments:
There are numerous syntax, spelling and grammatical errors throughout the manuscript.
The abstract and introduction content is weighed in favour of serpins, however, only AACT has been analysed. The content should be written to more accurately reflect the content of the manuscript and focus of the study.
In the introduction, the authors declare that "there is a paucity of information in the literature as to the relationship between COPD and protease-antiprotease imbalance". I do not agree with this statement given the number of published papers that have investigated profiles of a range of proteases and antiproteases in COPD patients.
Throughout the manuscript , the authors refer to Groups A - D, which they have notated based on GOLD guidelines. I believe this adds extra confusion to the manuscript and the results and I would prefer to see the patients grouped and notated according to the standard GOLD classifications, which are universally recognised.
Further explanation is needed as to what the authors mean by "low-risk" and "high-risk" groups and how this was interpreted.
Clarification is needed as to the method of analysis used to detect AACT, MMP-12, TIMP-1 and TIMP-2. "Activity" is mentioned throughout the text but it appears that only levels were detected by ELISA for all analytes.
The results within Table 2 would be much clearer if they were presented in graphical format. Also, the n numbers for all patients groups are very different to those presented in Table 1.
Presenting the correlation data either in tables or graphs would greatly improve the clarity of the results. It is unclear how correlations between analytes, FEV1 etc and exacerbations were analysed?
Did the authors analyse the MMP-9:TIMP-1 ratio?
A lot of focus in the discussion centres on the protease-antiprotease imbalance in COPD. The significance of this work (and any novel findings) and how it adds to the large body of data already published should be discussed. In addition, the limitations of the study should be acknowledged, particularly if levels and not activities of these proteins were analysed.